# An Electrochemical Immunosensor Based on SPA and rGO-PEI-Ag-Nf for the Detection of Arsanilic Acid

**DOI:** 10.3390/molecules27010172

**Published:** 2021-12-28

**Authors:** Yanwei Wang, Dongdong Ma, Gaiping Zhang, Xuannian Wang, Jingming Zhou, Yumei Chen, Xiaojuan You, Chao Liang, Yanhua Qi, Yuya Li, Aiping Wang

**Affiliations:** 1School of Life Science, Zhengzhou University, Zhengzhou 450000, China; W13271798101@126.com (Y.W.); 15225965189@163.com (D.M.); zhanggaip@126.com (G.Z.); zhjingming@126.com (J.Z.); yumeichen2012@163.com (Y.C.); youxiaojuan1988@163.com (X.Y.); chaonvlc@163.com (C.L.); yanhuaqi2007@163.com (Y.Q.); liyuya111@163.com (Y.L.); 2School of Life Science and Basic Medicine, Xinxiang University, Xinxiang 453003, China; wangxuannian@163.com

**Keywords:** arsanilic acid, polyethyleneimine, silver nanoparticles, reduced graphene oxide, staphylococcal protein A, electrochemical immunosensor

## Abstract

A sensitive electrochemical immunosensor was prepared for rapid detection of ASA based on arsanilic acid (ASA) monoclonal antibody with high affinity. In the preparation of nanomaterials, polyethyleneimine (PEI) improved the stability of the solution and acted as a reducing agent to generate reduced graphene oxide (rGO) with relatively strong conductivity, thereby promoting the transfer of electrons. The dual conductivity of rGO and silver nanoparticles (AgNPs) improved the sensitivity of the sensor. The synthesis of nanomaterials were confirmed by UV-Vis spectroscopy, X-ray diffraction, transmission electron microscopy and scanning electron microscopy. In the optimal experiment conditions, the sensor could achieve the detection range of 0.50–500 ng mL^−1^ and the limit of detection (LOD) of 0.38 ng mL^−1^ (S/N = 3). Moreover, the sensor exhibited excellent specificity and acceptable stability, suggesting that the proposed sensor possessed a good potential in ASA detection. Thus, the as-prepared biosensor may be a potential way for detecting other antibiotics in meat and animal-derived foods.

## 1. Introduction

Organoarsenicals (OAs) include arsanilic acid (ASA), roxarsone (ROX), carbarsone (CBA), and nitarsone (NPA), which served as feed additives in the 1950s; they not only can promote poultry weight gain and enhance meat pigmentation, but can also inhibit the growth and reproduction of harmful bacteria in the intestine [1,2,3]. Among them, ASA is widely used in animal production as food additives to control poultry diseases in China [4,5]. However, with the rapid development of national food safety detection technology, researchers found that ASA additives used at higher levels than recommended could result in toxic effects such as quadriplegia, ataxia, muscle tremors and blindness in the animal body [6,7]. The ASA that remained in animal tissues entered the body through the food chain, causing toxic and teratogenic effects. Therefore, the European and China issued a formal announcement to prohibit the use of ASA as a feed additive in livestock breeding [8,9,10]. In 2020, China stipulated that the maximum residue limit of ASA in pork and pork liver was 500 μg kg^−1^. Thus, an effective detection method is needed to avoid the abuse of such additives.

At present, the main detection methods of organic arsenic residues are high-performance liquid chromatography (HPLC) [11,12] and enzyme-linked immunosorbent assay (ELISA) [13]. However, the chromatographic detection methods require costly and sophisticated equipment, complex sample preparation, several organic reagents and professional instrument operators [14,15]. ELISA has the advantages of high sensitivity and high throughput, but it also presents the disadvantages of being time-consuming and theinability.to provide rapid onsite detection. Therefore, it is essential to develop a test method with low cost, easy operation and short time. Electrochemical immunosensor, a biosensor that combines electrochemical sensing system with immunoassay to detect antigens or antibodies in samples, [16] has attracted increasing attention because of its advantages of low energy consumption, simple equipment and easy miniaturization [17,18,19]. It is mainly applied in medical diagnosis, [20] environmental monitoring, [21] food safety monitoring, [22] etc. Therefore, it is a very promising detection technology. Currently, there is no report of electrochemical immunosensor for ASA detection. Thus, the purpose of this study is to prepare a label-free and sensitive electrochemical sensor to realize the detection of ASA.

Nanomaterials and conductive polymers have played a great role in biosensors and biofuel cells [23,24]. Graphene oxide (GO) as a graphene derivative, contains various functional groups and a large π-conjugated structure [25,26,27]. Due to its large specific surface area, it is often used to immobilize metal nanoparticles in the preparation of nanomaterials. Polyethyleneimine (PEI), with the large amount amine groups, can reduce and simultaneously functionalize GO under alkaline conditions and mild temperatures [28]. The conductivity of reduced graphene oxide (rGO) is eight times higher than that of GO [29], and its carboxyl group can be covalently bonded to PEI with highly active primary amino sites. Synthesized rGO-PEI has a larger specific surface and a bigger space charge layer than a two-dimensional sheet of graphene, so it exhibits excellent electrochemical activity [30]. Importantly, the appropriate stability of the rGO-PEI and its wide active surface area provide a more suitable matrix for the deposition of metal nanoparticles. AgNPs have a very low price, biocompatibility and superior conductivity, which are wildly used in electrochemical sensors. Ag+ can be reduced to AgNPs by glucose and in situ reduced to the surface of rGO-PEI, effectively preventing particles aggregation [31,32]. Nafion (Nf) can improve the dispersibility and film-forming ability of the solution [33,34]. Staphylococcal protein A (SPA) specifically recognizes and binds the Fc portion of the antibody, thus facilitating the fixation of highly targeted antibodies on the surface of the modified electrode without any antibody modification, resulting in a more effective antigen–antibody binding [35].

Therefore, based on the advantages of the above nanomaterials, we prepared rGO-PEI-Ag-Nf nanocomposites with large specific surface area and strong current signal, and established a detection method based on electrochemical immunosensor to realize the sensitive detection of ASA in pork and pork liver.

## 2. Materials and Methods

### 2.1. Reagents and Materials

Anti-ASA monoclonal antibody was prepared in the molecular immunology laboratory, Zhengzhou University. Graphene oxide (GO) was bought from Xianfeng Nanotechnology Co., Ltd. (Nanjing, China). Arsanilic acid (ASA), Roxarsone (ROX), Carbarsone (CBA), 3-amino-4-hydroxypheny arsonic acid (HAPA) were purchased from Dr. Ehrenstorfer GmbH (Augsburg, Germany). Nafion (Nf), NH_3_·H_2_O and bovine serum albumin (BSA) were acquired from Sigma-Aldrich (St. Louis, MO, USA). NaOH, Glucose and AgNO_3_ were bought from Sinopharm Chemical Reagent Co., Ltd. (Shanghai, China). PEI was purchased from Shanghai Macklin Biochemical Co., Ltd. (Shanghai, China). Staphylococcal Protein A (SPA) was acquired from Hangzhou NeuroPeptide Biological Science and Technology Incorporation, Ltd. (Hangzhou, China). Phosphate buffer solution (PBS) was prepared with a pH of 7.4. All reagents used were of analytical grade and the water was double distilled water.

### 2.2. Apparatus

CHI760E electrochemical workstation (Shanghai Chenhua Instrument Co., Ltd., Shanghai, China) was applied to carry out cyclic voltammetry (CV), differential pulse voltammetry (DPV) and electrochemical impedance spectroscopy (EIS). The electrochemical workstation consists of a three-electrode system, in which the working electrode was a gold electrode, the reference electrode was a saturated calomel electrode, and the counter electrode was a platinum wire electrode. Nanodrop 2000c UV-Vis spectrophotometer (Thermo Fisher Scientific Co., Ltd., St. Louis, MO, USA), X-ray diffractometer (X’Pert PRO, PANalytical, Almelo, The Netherlands), scanning electron microscope (Zeiss Sigma 300, Jena, Germany) and Talos L 120C transmission electron microscope (Thermo Fisher Scientific Co., Ltd., St. Louis, MO, USA) were used to characterize the morphological structures of the synthetic nanomaterials.

### 2.3. Synthesis of rGO-PEI and rGO-PEI-Ag Nanocomposites

The rGO-PEI-Ag nanocomposites were prepared according to the literature with a little change [36,37]. Figure 1 showed the basic synthesis process of nanomaterials. The specific synthesis process of rGO-PEI and rGO-PEI-Ag are described in the Appendix A.

### 2.4. The Assembly Process of the Electrochemical Sensor

Figure 1 showed the preparation process of immunosensor. First, the gold electrode was polished with 1 μm, 0.3 μm and 0.05 μm of Al_2_O_3_ powder respectively, and ultrasonically cleaned in water and anhydrous ethanol solution. The electrode surface was dried by nitrogen flow. Then, 9 μL rGO-PEI-Ag-Nf solution was dropped on the surface of the treated electrode and dried at room temperature. After that, 20 μL of SPA (0.3 mg mL^−1^) was added and incubated at 37 °C for 60 min. Next, 20 μL of anti-ASA antibody (1:1000) was incubated at 37 °C for 40 min. Subsequently, incubation was carried out with 20 μL of BSA (2.5 mg mL^−1^) for 30 min to block unbound active sites. After each step of modification, the electrode was cleaned with water to remove impurities resulting from physical absorption. Finally, the prepared BSA/anti-ASA/SPA/rGO-PEI-Ag-Nf sensor was immersed in PBS solution and stored at 4 °C for later use.

### 2.5. Electrochemical Measurements

CV was used to characterize the assembly process of the immunosensor in 10 mL PBS (0.1 M, PH 7.4), and the voltage rang was from −0.2 V to 0.6 V at a scan rate of 50 mV s^−1^. EIS was also used to characterize the assembly process in PBS containing 10 mL [Fe(CN)_6_]^3−/4−^ (5.0 mM) at frequency ranging from 0.1 Hz to 100 KHz, init E 0.226 V and amplitude 0.005 V. The response performance of the immunosensor was studied by DPV in PBS (0.1 M, pH 7.4) at a voltage from −0.08 to 0.26 V, amplitude of 0.05 V, pulse width of 0.05 s, incr E 0.004 V, pulse period 0.5 s and sampling width of 0.0167 s.

### 2.6. Pretreatment of Actual Samples

The pork and pork liver were bought from the local supermarket and pretreated according to the literature [38]. In short, 2 g samples were homogenized in 2 mL HCL (0.1 mol L^−1^) and treated with ultrasound for 8 min. Then 8 mL PBS (0.01mol L^−1^, pH = 7.4) was added and sonicated for 5 min. The mixture was centrifuged at 4 °C for 10 min to collect the supernatant and the pH was adjusted to 7.4.

## 3. Results and Discussion

### 3.1. Morphological Characterization of Nanocomposites

Firstly, we characterized GO, rGO-PEI and rGO-PEI-Ag by UV-visible spectra. Figure 2A showed that the characteristic absorption peak of GO appeared at 228 nm, which was due to the electronic π−π * transitions of C−C aromatic bonds in the spectrum of GO, and a little characteristic peak near 300 nm was likely related to the n−π * transitions of the C = O bond [39]. The peak associated with the π−π * transition at 228 nm was shifted to 258 nm in rGO-PEI, showing that rGO was successfully synthesized [36]. In rGO-PEI-Ag nanocomposites, besides the reduction peak at 258 nm, an absorption peak was observed at 420 nm, which was caused by the surface plasmon resonance of AgNPs [40]. X-ray diffraction patterns of nanomaterials were shown in Figure 2B. The diffraction peak with 4θ angles around 12.2°, which belonged to the diffraction peak of the (001) crystal plane of GO. Due to the reduction of PEI, the GO diffraction peak was disappeared in rGO-PEI and rGO-PEI-Ag. The four sharp diffracting peaks of as-synthesized AgNPs were located at 4θ angle of diffraction, 37.9°, 43.9°, 64.2° and 77.0° which were consistent with the (111), (200), (220) and (311) diffractions of face-centered cubic (FCC). These observations showed that rGO and PEI were combined together, and the synthesis of nanomaterials was feasible.

In order to further confirm the synthesis of nanocomposites, the ultrasonic treated GO and rGO-PEI-Ag solution were dripped onto copper meshes respectively, and observed by TEM. In Figure 2C, we could see that the GO nanosheets exhibited a wrinkled morphology. Figure 2D showed that minute AgNPs with diameters around 40 nm were stably distributed on the rGO-PEI sheets. These phenomena were in line with the previous literature [41]. SEM was also used to observe the appearance of rGO-PEI-Ag. Figure 3A showed that AgNPs were uniformly dispersed in the surface of rGO-PEI. The EDS results showed that C, N, O and Ag elements appeared in the nanocomposites, which proved that the rGO-PEI-Ag nanocomposites were formed in Figure 3C–F. These observation results suggested that PEI was successfully grafted onto rGO surface and the nanomaterials were successfully synthesized.

### 3.2. The Assembly Mechanism and Characterization of Electrochemical Immunosensor

At first, Nf had the excellent dispersion and film forming ability, thus it was used to obtain rGO-PEI-Ag-Nf solution, which was bounded to electrode. After that, the amino group of SPA could be assembled on the electrode surface due to the electrostatic interaction with the silver nanoparticles, and SPA can specifically recognize and bind the Fc part of the antibody by Vander Waals interaction, hydrogen binding and ionic binding [42]. Finally, using bovine serum albumin (BSA) to block the other unbound active sites, the sensor has been prepared and could be used for the detection of ASA.

The cyclic voltammetry (CV) were used to represent the assembly process of each material. As shown in Figure 4A, voltammogram a had no redox peak for the bare gold electrode since there was no redox substance. Voltammogram b showed a pair of obvious redox peak, mainly because silver nanoparticles could be used as redox probes, which had good electrical conductivity and enhanced electron transfer. After further addition of SPA, the voltammogram c was significantly reduced because the silver nanoparticles could bind with SPA and block electron transfer. SPA also had the function of directionally binding to the Fc fragment of antibody, so the peak current of voltammogram d decreased after the addition of anti-ASA. After adding BSA, voltammogram e dropped again. Finally, after the addition of ASA standard, the antigen and antibody specifically recognized and formed immune complex, which further blocked electron transfer and reduced the peak current (voltammogram f).

Figure 4B showed impedance values of different modified electrodes in PBS (pH 7.4) containing 5.0 mmol L^−1^ K_3_[Fe(CN)_6_]/K_4_[Fe(CN)_6_]. Charge transfer resistance (Ret) is the most sensitive and direct parameter that can be used to characterize events occurring at the working electrode surface. When different substances adsorb on the electrode surface, their values are different. The impedance value can be expressed by the diameter of the semicircle in EIS spectrum. The larger the diameter, the greater the impedance value, and the worse the corresponding conductivity. Compared with the impedance value of curve a (R_et_ = 194.5 Ω), a straight line without semicircle of curve b with the addition of rGO-PEI-Ag-Nf, indicating that the prepared nanocomposites had excellent electrical conductivity. The R_et_ of SPA (curve c), anti-ASA (curve d) and BSA (curve e) modified electrodes were gradually increased, 331.0 Ω, 571.1 Ω, 781.7 Ω, respectively, demonstrating that SPA, anti-ASA and BSA could the block electron transfer. The impedance value of the modified electrode (curve f) was further increased to 1150 Ω after the addition of ASA. The EIS results of different modified electrodes were corresponded to the CV, which indicated that the sensor had been successfully prepared.

In addition, we investigated the CV response of the sensor at different scanning speeds of 25–250 mV s^−1^. In Figure 5A, we could see that the peak current gradually increased with the increase of sweep speed. Figure 5B showed the oxidation peak current (Ipa) and reduction peak current (Ipc) were linear with the square root of sweep rate (v^1/2^). The linear regression equations were Ipa (μA) = 36.49X − 4.902 with a correlation coefficient of 0.9941 and Ipc (μA) = −35.94X + 64.16 with a correlation coefficient of 0.9970. The results showed that the redox process of immunosensor was controlled by diffusion [43].

### 3.3. Optimization of Experimental Conditions

To obtain the best experimental results, we optimized some key factors. We used CVs to optimize different experimental conditions according to the changing value of the current (Δ*I*). The concentration of AgNO_3_ had a large effect on the current signal, so we first optimized it. From Appendix A, we could conclude that when the concentration of AgNO_3_ was 0.12 M, the maximum current signal could reach about 620–650 μA. After that, with the increasing concentration of AgNO_3_, the current gradually decreased. This was because a large number of AgNPs accumulated on the rGO-PEI surface, and its dispersion decreased, which was not conducive to current transfer. Therefore, 0.12 M AgNO_3_ was chosen to prepare nanomaterials.

The SPA concentration and incubation time were optimized because SPA could bind to antibodies in a targeted way, thus influencing the binding amount of antigen and antibody later. As shown in Appendix A, we determined the best SPA concentration and incubation time by chessboard method. When 0.3 mg mL^−1^ of SPA was incubated for 60 min, Δ*I* reached the maximum value (281 μA). It showed that the SPA and the electrode have been fully integrated.

The antibody concentration and incubation time were also optimized by using the chessboard method in Appendix A. When the antibody concentration was 1:1000 and the incubation time was 40 min, Δ*I* attained 87 μA, suggesting that the binding of antibody reached saturation.

The binding time of the antigen was also a major factor influencing the performance of the immunosensor. Adding 20 μL ASA (20 ng mL^−1^) standard to the modified electrode. As shown in Appendix A, from 40 to 80 min, the value of Δ*I* increased with the increase of incubation time, and after 80 min, the value of Δ*I* began to decrease. Thus 80 min was selected as the optimal binding time of antigen and antibody.

Therefore, according to the optimal experimental conditions, we chose AgNO_3_ concentration of 0.12 M, incubation of 0.3 mg mL^−1^ SPA for 60 min, antibody concentration of 1:1000 incubation for 40 min, and the incubation time of antigen for 80 min in the following experiments.

### 3.4. Analysis Performance of Electrochemical Immunosensor for ASA Standard

A series of different concentrations of ASA standard (0.50–500 ng mL^−1^) were added to the surface of the modified electrode, and the reaction performance was studied under the optimal conditions. As shown in Figure 6A, the peak currents of DPV decreased gradually with the constant increase of ASA standard concentrations. Figure 6B showed that the current signal was proportional to the logarithm of the ASA concentrations, and the equation for the standard curve was Y = 25.29X + 27.18 with correlation coefficient R^2^ of 0.9952. The low detection limit of the electrochemical sensor was evaluated to be 0.38 ng mL^−1^ based on S/N = 3. Due to the large specific surface area and strong current signals of the synthesized rGO-PEI-Ag nanocomposites, the sensitivity of the sensor is improved. Compared with the previously reported detection method of ASA, [11,13,44,45] in Table 1, this method had a lower detection limit and a wider detection range.

### 3.5. The Reproducibility, Stability and Selectivity of the Immunosensor

Reproducibility, stability and selectivity were used to assess the electrochemical performance of the prepared immunosensor. Five gold electrodes were prepared in the same conditions for the determination of ASA samples. As shown in Figure 7A, the relative standard deviation among them was 4%, showing that the prepared sensor showed good reproducibility.

We continuously scanned the sensor in PBS solution for 50 cycles by CV and found that the current signal only dropped by 5% in Figure 7B. We immersed the prepared immunosensor in PBS solution and stored it in a refrigerator at 4 °C. The DPV signal was measured every four days, as shown in Figure 7C; the current signal dropped by 4% and 18% on the 8th and 28th days, respectively, indicating that the prepared sensor had good stability.

To evaluate the specificity of the immunosensor, we chose ASA structural analogues, roxarsone (ROX), carbarsone (CBA), 3-amino-4-hydroxypheny arsonic acid (HAPA), which were 5 times the concentration of the ASA. Figure 7D illustrated that all three interfering substances and the mixture did not cause large peak current changes. The DPV peak current of the mixture of ASA and other three interfering substances changed to only 2.3% of the peak current of the ASA solution, indicating that the immunosensor has good selectivity.

### 3.6. Analysis of Real Samples

A recovery experiment in the actual sample was used to assess the feasibility of the immunosensor. Pork and pork liver were bought from the supermarket and underwent the pretreatment as mentioned in Section 2.6. The ELISA method proved that the sample did not contain ASA. Different concentrations of ASA standard were added to samples of pork and pork liver, and the result is shown in Table 2. The recovery rate added to the pork sample was 97.73–102.9%, and the RSD was 1.1–1.5%. The recovery rate in the pork liver sample was 95–103%, and RSD was 1.1–4.7%. The results showed that the prepared sensor had high accuracy and could be used for the detection of animal-derived food.

## 4. Conclusions

In this work, we successfully prepared a high-sensitivity and label-free electrochemical immunosensor for the detection of ASA using rGO-PEI-Ag-Nf nanocomposites and SPA. In the preparation of nanocomposites, PEI could improve the stability of the solution. Moreover, it could also be used as a reducing agent for GO and combined with the carboxyl group of rGO to form a more stable matrix for the immobilization of AgNPs. For the first time, the electrochemical detection of ASA was realized. The desired nano-sensor showed a linear dynamic range of 0.50 to 500 ng mL^−1^ and low limit of detection (LOD) as 0.38 ng mL^−1^ in pork and pork liver sample, which revealed excellent analytical performance for the detection of ASA with high sensitivity and reproducibility. Therefore, it could be a promising detection method for other antibacterial substances detection.

## Data Availability

Not applicable.

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
