# Peer review of "An Electrochemical Immunosensor Based on SPA and rGO-PEI-Ag-Nf for the Detection of Arsanilic Acid"

_molecules, 2021, doi:10.3390/molecules27010172_

Round 1
Reviewer 1 Report
Reviewer report on Manuscript Draft ‘An electrochemical immunosensor based on SPA and rGO-PEI-Ag-Nf for the detection of arsanilic acid’
In this work, authors designed a high-sensitivity and label-free electrochemical immunosensor for the detection of arsanilic acid using rGO-PEI-Ag-Nf nanocomposites. For the first time, the electrochemical detection of arsanilic acid was realized, what is especially interesting, because arsanilic acid is a small molecule, which mostly is hardly detectable using antibodies.
This manuscript is in the scope of journal is rather well written and interestingly addressed. Therefore, the manuscript can be published after some minor improvements and corrections:
The arsanilic acid is a small molecule, which mostly is hardly detectable using antibodies, therefore, the antibody development procedure (from producer) and some proofs that it interacts with arsanilic acid are required.
It will be nice to organize the Introduction in such way that the last paragraph in few sentences will clearly address the main aim of the manuscript.
Recent advances, insights and reviews on the development of electrochemical immunosensors (Conducting Polymers in the Design of Biosensors and Biofuel Cells. Polymers 2021, 13, 49.) and in some other affinity sensors (Advances in molecularly imprinted polymers based affinity sensors (Review). Polymers 2021, 13, 974.) based on semiconducting structures/layers from MDPI journals could be overviewed and discussed.
Figure 4 represents Nyquist plot of electrochemical impedance spectroscopy (EIS) data. But it should be noted that representation of Nyquist plot is correct when both scales of X and Y are axis are the same, then it allows visually assess presented data and visually determine Rct and some other characteristics of evaluated electrochemical system. In recent graph Y-axis is significantly ‘extended’ in comparison to X-axis, therefore, corresponding correction is needed.
Electrochemical impedance spectroscopy (EIS) data should be evaluated using the most suitable equivalent circuit and compared and discussed with data on the application of electrochemical impedance spectroscopy (EIS) based immunosensors.
Author Response
Dear reviewer:
We deeply appreciate your positive and constructive comments and suggestions concerning our manuscript “An electrochemical immunosensor based on SPA and rGO-PEI-Ag-Nf for the detection of arsanilic acid”. The revised manuscript has been uploaded again, which we would like to submit for your kind consideration. We have studied comments carefully and tried our best to make correction and some changes through “Track Changes” in our manuscript.
Comment 1: The arsanilic acid is a small molecule, which mostly is hardly detectable using antibodies, therefore, the antibody development procedure (from producer) and some proofs that it interacts with arsanilic acid are required.
Response: Thank you very much for your review and suggestion. Anti-ASA monoclonal antibody was prepared in the molecular immunology laboratory, Zhengzhou University. The main preparation process was as follows: The small molecule antigen of arsanilic acid had only immunoreactivity and no immunogenicity. Therefore, it needed to be combined with macromolecular proteins (BSA) to induce the immune response to hapten. The mice were immunized with ASA-BSA complete antigen, and the hybridoma cells were obtained by cell fusion after the fourth immunization. Then the hybridoma cell line 4A4 with strong positive and high sensitivity was injected into mice to produce ascites, and then purified by bitter-ammonium sulfate law. The antibody titer was 1:204800, the IC50 was 0.757µg mL-1 and the affinity constant was 2.34 x 108 L moL-1. The hapten determinants of small molecules contain a binding site with antibodies, which can react with antibodies and have immune reactivity. Therefore, the detection of small molecules can be realized by monoclonal antibodies. In the mouse serum sensitivity test, we also proved that the prepared antibody could bind to small molecule antigen by indirect competitive ELISA.
Comment 2: It will be nice to organize the Introduction in such way that the last paragraph in few sentences will clearly address the main aim of the manuscript.
Response: Thank you very much for this reminder. In the last paragraph of introduction, we added “Therefore, based on the advantages of the above nanomaterials, we prepared rGO-PEI-Ag-Nf nanocomposites with large specific surface area and strong current signal, and established a detection method based on electrochemical immunosensor to realize the sensitive detection of ASA in pork and pork liver.” (page 2 line 59-61)
Comment 3: Recent advances, insights and reviews on the development of electrochemical immunosensors (Conducting Polymers in the Design of Biosensors and Biofuel Cells. Polymers 2021, 13, 49.) and in some other affinity sensors (Advances in molecularly imprinted polymers based affinity sensors (Review). Polymers 2021, 13, 974.) based on semiconducting structures/layers from MDPI journals could be overviewed and discussed.
Response: Thank you very much for your review and suggestion. (Conducting Polymers in the Design of Biosensors and Biofuel Cells. Polymers 2021, 13, 49.) and (Advances in molecularly imprinted polymers based affinity sensors (Review). Polymers 2021, 13, 974.) were cited in the first sentence of the third paragraph of the introduction. (page 2 line 44)
Comment 4: Figure 4 represents Nyquist plot of electrochemical impedance spectroscopy (EIS) data. But it should be noted that representation of Nyquist plot is correct when both scales of X and Y are axis are the same, then it allows visually assess presented data and visually determine Rct and some other characteristics of evaluated electrochemical system. In recent graph Y-axis is significantly ‘extended’ in comparison to X-axis, therefore, corresponding correction is needed. Electrochemical impedance spectroscopy (EIS) data should be evaluated using the most suitable equivalent circuit and compared and discussed with data on the application of electrochemical impedance spectroscopy (EIS) based immunosensors.
Response: Thank you very much for your suggestions. We adjusted the ratio of X-axis and Y-axis in Figure 4 (B). Because the data of X-axis and Y-axis are very different, we enlarged the Y-axis to a certain scale to make the picture look more intuitive and clear. There are no strict requirements for the scale of x-axis and Y-axis of EIS in different articles, for example, “A high sensitive label-free electrochemical immunosensor based on poly (indole-5carboxylicacid) with ultra-high redox stability”. We supplemented the equivalent circuit diagram in Figure 4 (B). The EIS data were fitted to a Randles equivalent circuit, containing charge transfer resistance (Ret), electrolyte resistance (Rs), constant phase element (CPE), and Warburg element (ZW). Rs and Zw are hardly affected by the modification of the electrode surface, while the change of CPE is not obvious compared to the change of Ret. Thus, Ret is the most sensitive and direct parameter that can be used to characterize events occurring at the working electrode surface. When different substances adsorb on the electrode surface, the value of Ret will change. In the revised manuscript, we added “Charge transfer resistance (Ret) is the most sensitive and direct parameter that can be used to characterize events occurring at the working electrode surface. When different substances adsorb on the electrode surface, their values are different.” and added Ret values of the same electrode at different stages. (page 6 line 172-174, line176-180).
Reviewer 2 Report
In this study, authors report the first label-free electrochemical immunosensor for the detection of arsanilic acid in meat.
The work is relevant and interesting. Also, it is in the scope of the journal.
However, There are some issues that must be clarified.
Introduction
- Lines 47, 48, 49 – The sentences can be merged.
- Line 59 - Specify "actual samples".
Material and Methods
- Line 86 - Remove the dot between” Scheme 1”. Same for Line 192 “Table 1”. In the same line, authors must specify the electrode used throughout the work (gold electrode).
- Line 91 - The BSA amount or concentration must be added.
- In section 2.5. the authors must provide information regarding the reference and counter/auxiliary electrodes employed in the work, as well as the volume of PBS and/or [Fe(CN)6]3-/4- used for electrochemical measurements.
Also, in line 147 it is mentioned “containing 5.0 m mol L-1 147 K3[Fe(CN)6]/K4[Fe(CN)6]”. “m” should be “mL” or should be removed? Please clarify.
- Section 2.6. Are samples analysed immediately after pretreatment? What is the aliquot volume used?
Results and Discussion
- Lines 138-145 - The graphical representation should be referred to as a voltammogram and not as a "curve".
- Figure 4 - The image is correctly subtitled; however, the colors of the voltammograms (a-f) can be the same for both (A) and (B) to aid visual interpretation of the same parameter.
- Since the authors investigated the CV response of the sensor at different scanning speeds (lines 156-161), sensitivity must be calculated. Give quantitative data on electrochemical sensitivity. The surface area can be calculated from electrochemical measurements via the Randle-Sevcik equation.
- Section 3.3.
- In the electrochemical morphological characterization (3.1.) authors report that CV and EIS techniques were used. However, in the optimization of experimental conditions (section 3.3.) no electrochemical technique was mentioned. Although authors refer to in Line 98, the technique can be included in the text / Figure and/or Table legends.
- Table S2 - correct the word “antiboby”.
- In the optimization process, what is the concentration and volume of antigen?
- Although â–³ I is referred to, authors should clarify the obtained electrochemical signal. For example, (lines 176-177) in the sentence “When 0.3 mg mL-1 of SPA was incubated for 60 minutes, â–³ I reached the maximum” authors must conclude with the measured parameter (maybe the maximum current intensity difference?).
- Additionally, the legend of Table S1 should be improved. Authors must clarify the unit of the obtained results as well as the employed electrochemical technique. Same comment for Table S2.
- Once the time was optimized, the authors could conclude the section 3.3. referring to the total assay time of the developed immunosensor.
- In Figure 6 (B) the legend mention “error bars: SD, n=3”. Thus, authors must provide the error bars in the figure.
- Section 3.4. (lines 191-192). Nanomaterials have been extensively characterized in this work; thus, its main features should be highlighted to reinforce their uniqueness. Authors should discuss their main advantages to be used in immunosensors, compared to the works mentioned in the table 1.
Author Response
Dear reviewer:
We deeply appreciate your positive and constructive comments and suggestions concerning our manuscript “An electrochemical immunosensor based on SPA and rGO-PEI-Ag-Nf for the detection of arsanilic acid”. The revised manuscript has been uploaded again, which we would like to submit for your kind consideration. We have studied comments carefully and tried our best to make correction and some changes through “Track Changes” in our manuscript.
Comment 1: Lines 47, 48, 49 – The sentences can be merged.
Response: Thank you very much for your suggestions. Lines 47, 48, 49 was merged into “The conductivity of reduced graphene oxide (rGO) is 8 times higher than that of GO, and its carboxyl group can be covalently bonded to PEI with highly active primary amino sites.” in the revised manuscript. (page 2 line 52-53)
Comment 2: Line 59 - Specify "actual samples".
Response: Thank you very much for your suggestions. The “actual samples” was changed to “pork and pork liver”. (page 2 line 65)
Comment 3: Line 86 - Remove the dot between” Scheme 1”. Same for Line 192 “Table 1”. In the same line, authors must specify the electrode used throughout the work (gold electrode).
Response: Thank you very much for your suggestions. The dot between scheme 1 and table 1 in the article has been deleted. (page 4 line 105, page 8 line 235) In each part of the manuscript, we illustrated that the electrodes used were gold electrodes, for example, we modified “electrodes” to “gold electrodes” in section 2.4. (page 4 line 105)
Comment 4: Line 91 - The BSA amount or concentration must be added.
Response: Thank you very much for your reminder. I'm very sorry that the concentration of BSA was not indicated in detail due to our negligence. During the assembly of the electrochemical sensor, the concentration of BSA was 2.5 mg mL-1. (page 4 line 110)
Comment 5: In section 2.5. the authors must provide information regarding the reference and counter/auxiliary electrodes employed in the work, as well as the volume of PBS and/or [Fe(CN)6]3-/4- used for electrochemical measurements.
Also, in line 147 it is mentioned “containing 5.0 m mol L-1 K3[Fe(CN)6]/K4[Fe(CN)6]”. “m” should be “mL” or should be removed? Please clarify.
Response: Thank you for your good suggestion. We added that “the electrochemical workstation consists of a three-electrode system, in which the working electrode was a gold electrode, the reference electrode was a saturated calomel electrode, and the counter electrode was a platinum wire electrode”. (page 2 line 79-81) And the volume of PBS and [Fe(CN)6]3-/4-were both 10 mL in electrochemical measurements in our revised manuscript. (page 4 line 114,116)
We are so sorry for this error and thank you for pointing it out. The concentration unit of K3[Fe(CN)6]/K4[Fe(CN)6] was mmol L-1. (page 6 line 171)
Comment 6: Section 2.6. Are samples analysed immediately after pretreatment? What is the aliquot volume used?
Response: Thank you for your comment. The treated pork and pig liver samples can be directly added with the standard of arsanilic acid for standard addition and recovery experiment, or can be frozen at –20℃, and the standard can be added for detection when in use. We diluted the obtained pork and pig liver samples 30 times before adding ASA standard to minimize the residue of the sample itself. The obtained pork and pig liver samples with a certain concentration of ASA standard were added with 20 μL dropwise on the prepared electrode to detect the electrochemical signal.
Comment 7: Lines 138-145 - The graphical representation should be referred to as a voltammogram and not as a "curve".
Response: Thank you very much for this reminder. In the revised manuscript, we changed “curve” to “voltammogram”. (page 6 line 163-170)
Comment 8: Figure 4 - The image is correctly subtitled; however, the colors of the voltammograms (a-f) can be the same for both (A) and (B) to aid visual interpretation of the same parameter.
Response: Thank you very much for your review and suggestion. In the revised manuscript, the colors of the voltammograms (a-f) of Figures (A) and (B) were revised to be the same. (page 6 figure 4)
Comment 9: Since the authors investigated the CV response of the sensor at different scanning speeds (lines 156-161), sensitivity must be calculated. Give quantitative data on electrochemical sensitivity.
The surface area can be calculated from electrochemical measurements via the Randle-Sevcik equation.
Response: Thank you very much for this reminder. At different sweep speeds, the sensitivity of this sensor can be obtained as 36.49 in the linear relationship between the square root of the sweep speed and the corresponding current. And in the nanocomposites we prepared, silver nanoparticles were fixed on the electrode surface as redox substances. Through Randle Sevcik formula, it could be concluded that the active surface area of the electrochemical sensor had little difference at different scanning rates, which was 0.0035 cm2.
Comment 10: Section 3.3.
- In the electrochemical morphological characterization (3.1.) authors report that CV and EIS techniques were used. However, in the optimization of experimental conditions (section 3.3.) no electrochemical technique was mentioned. Although authors refer to in Line 98, the technique can be included in the text / Figure and/or Table legends.
Response: Thank you very much for this reminder. In section 3.3, for the optimization of experimental conditions, we mainly use cyclic voltammetry. Therefore, in the revised manuscript, added "We used CVs to optimize different experimental conditions according to the current change value (â–³â… )" for supplementary explanation. (page 7 line 205-206)
And in the modified supporting materials, we also supplemented the electrochemical test method in the legends of table S1 and table S2.
- Table S2 - correct the word “antiboby”.
Response: We are so sorry for this error and thank you for pointing it out. We have made a correction in “Table S2” in the supporting information.
- In the optimization process, what is the concentration and volume of antigen?
Response: Thank you very much for this reminder. The concentration of antigen was 20 ng mL-1 and the volume was 20 μL. In the revised manuscript, supplemented “Adding 20 μL ASA (20 ng mL-1) standard to the modified electrode.” (page 7 line 219-220)
- Although â–³ I is referred to, authors should clarify the obtained electrochemical signal. For example, (lines 176-177) in the sentence “When 0.3 mg mL-1 of SPA was incubated for 60 minutes, â–³ I reached the maximum” authors must conclude with the measured parameter (maybe the maximum current intensity difference?).
Response: Thank you very much for your good suggestion and we fully agree with you. Therefore, we listed the specific values of current intensity changes â–³ I (281 μA) of SPA at different concentrations and incubation times. And the maximum current change value of antibody at different concentrations and incubation timeâ–³ I (87 μA). (page 7 line 215, 217)
- Additionally, the legend of Table S1 should be improved. Authors must clarify the unit of the obtained results as well as the employed electrochemical technique. Same comment for Table S2.
Response: Thank you very much for this reminder. The legend of table S1 was changed to “Table S1 Optimization of SPA concentration and incubation time by CVs. The data in the table was the variation of the intensity of the current (μA).”, and the legend of table S1 was changed to “Table S2 Optimization of antibody concentration and incubation time by CVs. The data in the table was the variation of the intensity of the current (μA).” in the revised supporting information.
Once the time was optimized, the authors could conclude the section 3.3. referring to the total assay time of the developed immunosensor.
Response: Thank you very much for your review and suggestion. At the end of section 3.3, we added “Therefore, according to the optimal experimental conditions, we chose AgNO3 concentration of 0.12 M, incubation of 0.3 mg mL-1 SPA for 60 minutes, antibody concentration of 1:1000 incubation for 40 minutes, and the incubation time of antigen for 80 minutes in the following experiments.” (page 7 line 223-225)
Comment 11: In Figure 6 (B) the legend mention “error bars: SD, n=3”. Thus, authors must provide the error bars in the figure.
Response: We are so sorry for this error and thank you for pointing it out. We modified figure 6 (B) and added error bars to the standard curve. (page 8 figure 6)
Comment 12: Section 3.4. (lines 191-192). Nanomaterials have been extensively characterized in this work; thus, its main features should be highlighted to reinforce their uniqueness. Authors should discuss their main advantages to be used in immunosensors, compared to the works mentioned in the table 1.
Response: Thank you very much for your review and suggestion. In section 3.4, we added “Due to the large specific surface area and strong current signals of the synthesized rGO-PEI-Ag nanocomposites, the sensitivity of the sensor is improved.” in the revised manuscript. (page 8 line 233-234).
Round 2
Reviewer 1 Report
Corrections were performed, the manuscript has been improved and now it is suitable for publishing.